# ΦCrAss001 represents the most abundant bacteriophage family in the human gut and infects *Bacteroides intestinalis*

Andrey N. Shkoporov [1], Ekaterina V. Khokhlova[1], C. Brian Fitzgerald[1], Stephen R. Stockdale[2], Lorraine A. Draper[1], R. Paul Ross[1,2,3] & Colin Hill[1,4]

CrAssphages are an extensive and ubiquitous family of tailed bacteriophages, predicted to infect bacteria of the order *Bacteroidales*. Despite being found in ~50% of individuals and representing up to 90% of human gut viromes, members of this viral family have never been isolated in culture and remain understudied. Here, we report the isolation of a CrAssphage (ΦCrAss001) from human faecal material. This bacteriophage infects the human gut symbiont *Bacteroides intestinalis*, confirming previous in silico predictions of the likely host. DNA sequencing demonstrates that the bacteriophage genome is circular, 102 kb in size, and has unusual structural traits. In addition, electron microscopy confirms that ΦcrAss001 has a podovirus-like morphology. Despite the absence of obvious lysogeny genes, ΦcrAss001 replicates in a way that does not disrupt proliferation of the host bacterium, and is able to maintain itself in continuous host culture during several weeks.

[1] APC Microbiome Ireland, University College Cork, Cork T12 YT20, Ireland. [2] Department of Food Biosciences, Teagasc Food Research Centre Moorepark, Fermoy P61 C996, Ireland. [3] College of Science, Engineering and Food Science, University College Cork, Cork T12 YT20, Ireland. [4] School of Microbiology, University College Cork, Cork T12 YT20, Ireland. Correspondence and requests for materials should be addressed to A.N.S. (email: andrey.shkoporov@ucc.ie) or to C.H. (email: c.hill@ucc.ie)

The human gut virome contains a vast number of bacterial, mammalian, plant, fungal and archaeal viruses[1–3]. There is a growing body of evidence supporting specific and consistent alterations of the gut virome in a number of human diseases and conditions, including inflammatory bowel disease (IBD), malnutrition and AIDS[4–6]. Currently, our understanding of the physiological significance of the human gut virome is limited by the fact that the vast majority of these viruses cannot be taxonomically classified or linked to any particular hosts (so-called viral dark matter[7]).

A study by Dutilh et al.[8] demonstrated the presence in ~50% of human samples of highly abundant sequences, which when cross-assembled from multiple sources, indicated the presence of a unique bacteriophage they named crAssphage (cross Assembly). The fully assembled 97 kb DNA genome showed no homology to any known virus, even though in some subjects it completely dominated the gut virome (up to 90% of sequencing reads). Indirect evidence suggested that Bacteroides species is the likely host[8]. A later study, based on detailed sequence analysis of proteins encoded by this virus, predicted Podoviridae-like morphology and placed it into the novel diverse and expansive family-level phylogenetic group of loosely related bacteriophages termed crAss-like bacteriophages[9]. Genomes of members of this group are highly abundant in the human gut and are also present in other diverse habitats including the termite gut, terrestrial/groundwater and the oceans[9–12]. However, in the absence of a known host, no member of this family has been isolated and nothing is known of the biological properties of these crAss-like viruses from the human gut.

In a recent analysis[13], we de novo-assembled 244 genomes of crAss-like bacteriophages from the human gut and classified them into 10 genus- and 4 subfamily-level taxonomic groups (Alphacrassvirinae, Betacrassvirinae, Gammacrassvirinae and Deltacrassvirinae) based on percentage of shared orthologous genes. As a result of this study, we can state that 98–100% of healthy adults from Western cohorts carry at least one or more types of these bacteriophages, albeit with widely varying relative abundances.

Here we report on the first successful isolation of a crAssphage on a single host and describe its key biological properties. Based on electron microscopy of propagated crAss001, we confirm that crAss-like bacteriophages possess podovirus-like morphology. We also demonstrate the ability of the virus to stably co-replicate with its Bacteroides intestinalis host in equilibrium for many generations in vitro, which mimics earlier observed ability of crAss-like bacteriophages to maintain stable colonisation of the mammalian gut[13,14].

## Results

**Isolation and genome analysis of ΦcrAss001.** The replication strategy of crAss-like bacteriophages is unknown and all attempts to use standard plaque assays on semi-solid agar have failed. We therefore attempted to detect crAss-like bacteriophage replication using a broth enrichment strategy. Phage-enriched filtrates of faecal samples were collected from 20 healthy adult Irish volunteers, pooled and used to infect pure cultures of 54 bacterial strains representative of the commensal human gut microbiota (Supplementary Data 1). After three successive rounds of enrichment, cell-free supernatants for each of the 54 strains were subjected to shotgun metagenomic sequencing. Analysis of the assembled sequencing reads demonstrated that the supernatant from the strain B. intestinalis APC919/174 was dominated by a single 102.7 kb contig (~98% of reads) related to a known but previously uncultured crAss-like bacteriophage, IAS virus, identified from the human gut[9,15].

The genome of bacteriophage ΦcrAss001 is 102,679 bp (GenBank MH675552) and is circular or circularly permuted. We identified 105 protein-coding genes (open reading frames (ORFs) of length 907239 bp) and 25 tRNA genes specific for 17 different amino acids. Cumulative G + C content of the bacteriophage genome is 34.7 mol%, which is significantly lower than that of APC919/174 (42.5 mol%, GenBank QRES00000000) and other published B. intestinalis genomes (42.843.5 mol%). An interesting structural feature of the genome is its apparent division into two parts of roughly equal size with strictly opposite gene orientation and inverted GC skew, possibly reflecting the direction of transcription and/or replication (Fig. 1). Analysis of local read coverage revealed spikes of elevated sequence coverage consistent with the direct terminal repeat (DTR) type of genome packaging[16,17] with a 576-bp redundant sequence predicted to serve as the DTR.

Functional gene annotation was performed using a comprehensive approach, which included BLASTp amino-acid sequence homology searches against NCBI nt database, hidden Markov model (HMM) searches against UniProtKB/TrEMBL database[18] and profile-profile HMM searches with HHpred against PDB, PFAM, NCBI-CDD and TIGRFAM databases[19,20]. This allowed for the functional annotation of 57 genes and assignment of a further 11 genes to conserved protein families with unknown functions (Supplementary Data 2). Ten identifiable structural protein genes: major capsid protein (MCP), tail and tail appendage proteins, as well as three genes responsible for lytic functions were clustered on the right-hand side of the genome, suggesting that the remaining un-annotated genes in this part of the genome may also be responsible for structure and assembly of viral particles, as well as cell lysis. By contrast, the left-hand side predominantly harboured genes involved in replication, recombination, transcription and nucleotide metabolism. Putative DNA-binding and transcriptional regulation proteins were located in the proximal portions of the two oppositely oriented genome halves, suggesting their role in governing the transcription of gene modules located downstream on both sides of the genome (Fig. 1). Two genes (gp3, β-fructosidase; and gp5, ferredoxin-thioredoxin reductase) were predicted to be involved in auxiliary metabolic processes, in that they are unrelated with bacteriophage replication and virion assembly. No identifiable lysogeny module or integrase gene were found.

**Structure of ΦcrAss001 virions.** Comparison of ΦcrAss001 MCP using HMM-HMM searches with other bacteriophage MCPs with experimentally solved tertiary structure revealed a significant match in the C-terminal region (aa 176405) with the MCP of Staphylococcus phage 812 (PDB 5LII [https://www.rcsb.org/structure/5LII]). This enabled us to perform homology modelling, which revealed a typical HK97-like fold. Very similar tertiary structures were predicted for MCPs of other members of the family: prototypical crAssphage (p-crAssphage) and IAS virus (Fig. 2). Matrix-assisted laser desorption/ionization mass spectrometry (MALDI-TOF) analysis of virion proteins separated by SDS-polyacrylamide gel electrophoresis (SDS-PAGE) identified the presence of most of the predicted structural proteins, with exception of the putative head fibre protein (gp21), the C-type lectin (gp24) and the tail tubular protein (gp43). It is possible that these gene products could not be detected due to their extremely low copy number or, due to mis-annotation of the corresponding ORFs. Additional putative structural proteins of unknown function, not identified by genome analysis, included gp29 and gp36. Unexpectedly, three high-molecular-weight subunits of bacteriophage RNA polymerase[9] (gene products 47, 49 and 50) were also detected as a putative virion-associated proteins (Fig. 3a).

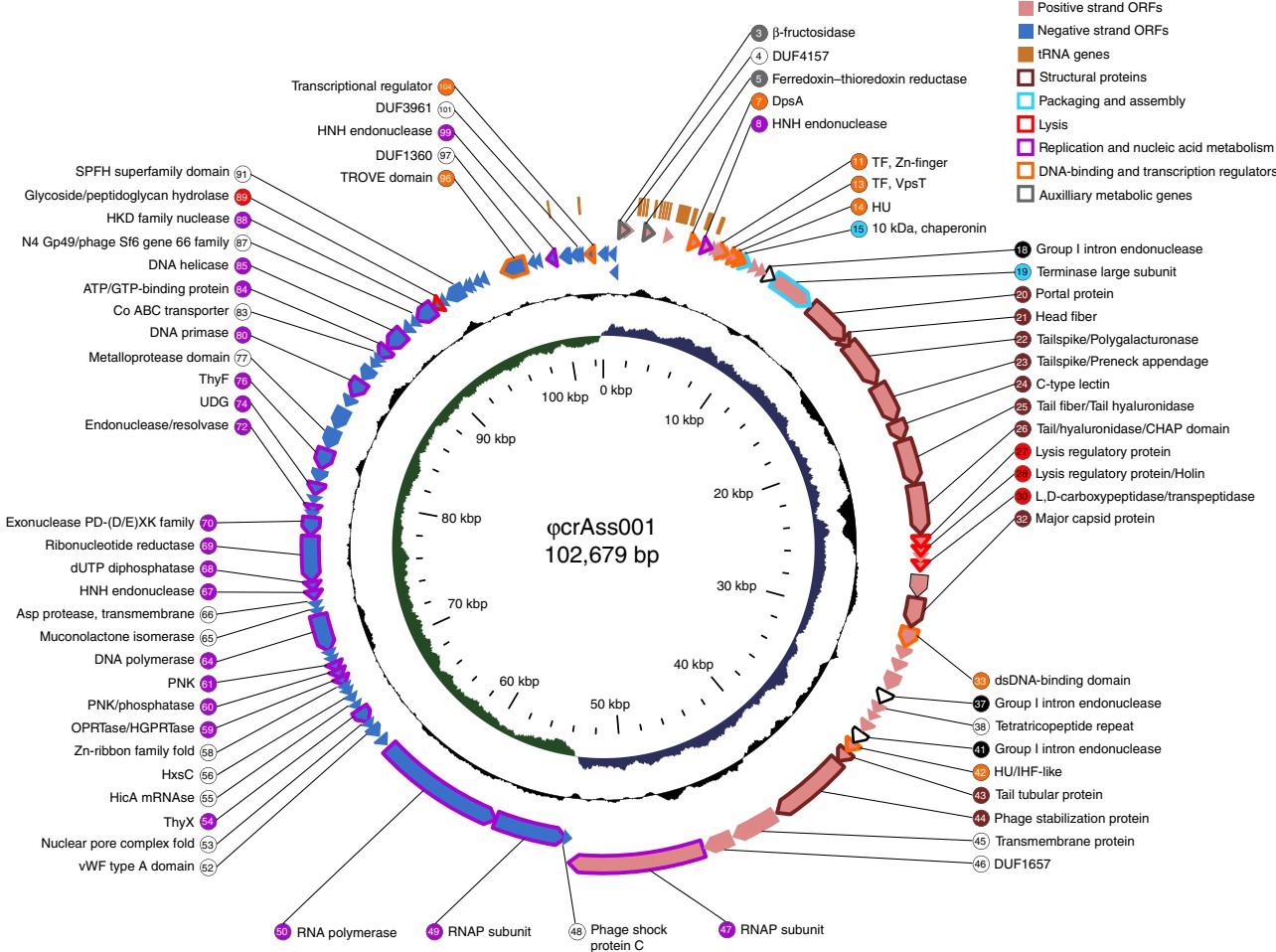

**Fig. 1** Circular map of ΦcrAss001 genome. Inner circle (green and blue), GC skew; middle circle (black), G + C content; outer circle (pink and purple fill), protein-coding genes (ORFs); outermost circle (orange fill), tRNA genes. Stroke colour on ORFs and fill colour on gene numbers corresponds with the general predicted function (see colour legend for details); black stroke, introns; genes with no functional annotations (coding for hypothetical proteins) are left unlabelled

Moderate levels of discrepancy between predicted molecular weights of virion proteins and observed molecular weight could be attributed to post-translational modifications or anomalous electrophoretic mobility of the polypeptides.

Transmission electron microscopy (TEM) of ΦcrAss001 virions revealed a podovirus-like morphology (Fig. 3b). Virion heads are isometric with a diameter of 77.2 ± 3.3 nm (mean ± SD). Tails are 36.1 ± 3.6 nm long with elaborate structural features and several side appendages of variable length. This is in agreement with previous TEM observations of podoviruses of similar dimensions (~76.5 nm in diameter) isolated from a faecal sample rich in a mixture of several highly prevalent crAss-like bacteriophages[13]. However, the predominant component of that mixture, Fferm_ms_2 (Fig. 3c), belongs to a different subfamily of crAss-like viruses, and demonstrates a markedly different structure of the predicted tail region in the genome (Supplementary Data 3). This could explain the difference in TEM tail appearance between ΦcrAss001 and Fferm_ms_2, with the latter supposedly demonstrating a much simpler T7-like tail structure.

**Taxonomic position of ΦcrAss001 with related bacteriophages.** According to the recently proposed classification scheme based on functional gene repertoire and protein sequence homology[9], ΦcrAss001 fits into the candidate genus VI, subfamily *Betacrassvirinae* (IAS virus[15] subgroup) of crAss-like bacteriophages

(Fig. 3c). Our recent analysis identified a common presence of similar uncultured bacteriophages in the gut microbiotas of healthy Irish and US adults, Irish elderly people, as well as in healthy Irish infants and healthy and malnourished Malawian infants[13], where they were detectable in 62, 44, 50, 10, 14 and 16% of cases, respectively, and in some cases represented up to 61% of virome reads (Supplementary Figure 1).

When genomes of ΦcrAss001, IAS virus and other related putative bacteriophages were compared, most of the sequence variability (including variation in the number and size of ORFs) was concentrated in a region putatively coding for tail spike/tail fibre subunits (gene products 22, 23, 25 and 26; Fig. 3c). This suggests a high level of variability in receptor-binding proteins and potentially high levels of host specialisation amongst crAss-like viruses. Interestingly, three large ORFs predicted to code for RNA polymerase subunits and occupying the medial portion of the ΦcrAss001 genome are located on different coding strands. Similar arrangements could also be seen in other candidate genus VI genomes (Fig. 3c), which differs from that observed for p-crAssphage and opens interesting regulatory opportunities.

**Biological properties of ΦcrAss001.** ΦcrAss001 could be effectively propagated in vitro using standard techniques using an exopolysaccharide -producing *B. intestinalis* 919/174 as its host. It was able to form readily visible plaques in agar overlays (Fig. 3d)

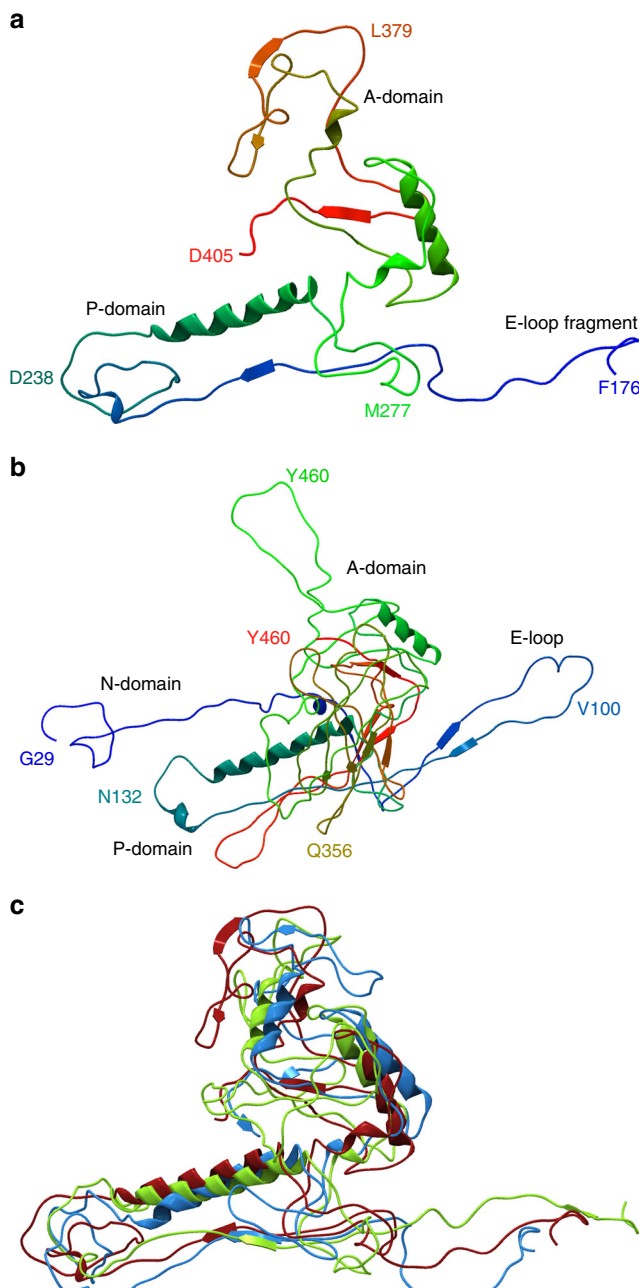

**Fig. 2** Predicted tertiary structure of ΦcrAss001 major capsid protein (MCP). **a** Homology-based modelling of aa 176405 fragment of ΦcrAss001 MCP. A-domain, axial domain; P-domain, peripheral domain; N-terminal domain and E-loop could not be resolved due to lack of similarity with the used template; colour gradient reflects aa position in the sequence (blue to green to red from N terminus to C terminus). **b** *Staphylococcus* phage 812 MCP (PDB 5LII [https://www.rcsb.org/structure/5LII][40]) used as template in HHPred-MODELLER suit (HHPred probability score 96.87%, e-value 0.00081). **c** Superposition of φcrAss001 MCP model (red) with p-crAssphage (green) and IAS virus (blue) MCP models (probability scores against PDB 5LII [https://www.rcsb.org/structure/5LII] 99.37, 96.89%, e-values 4e-14, 0.00063, respectively)

of uncultured crAss-like bacteriophages detected in the human gut[13], together with variability of their predicted receptor-binding genes and notorious resistance to isolation in culture suggest that crAss-like viruses in general and ΦcrAss001 in particular are likely to be narrow specialists, rather than generalists in terms of their host range. However, the large difference in GC content between ΦcrAss001 and its only known host may suggest that the bacteriophage did not evolve on this particular host, and/or it may infect other hosts in its natural habitat, where rapid host switching via recombination within the enormously rich gene pool can occur[21,22]. A BLASTn search of CRISPR spacer sequences and tRNA genes predicted from 5591 draft and complete bacterial genomes, included into NCBI RefSeq database of prokaryotic genomes, against the φcrAss001 genome did not reveal any clues of alternative hosts (see supplementary methods). Interestingly, however, most of the top CRISPR spacer hits were against taxonomically diverse species associated with the human gut and corresponded to putative selfish genetic elements in ΦcrAss001 (Supplementary Figure 2). Neither APC919/174 nor the other three strains of *B. intestinalis* in RefSeq possessed any CRISPR arrays.

In a one-step growth experiment with multiplicity of infection (MOI) of ~1 ΦcrAss001 demonstrated a long latent period of 120 min that was followed with a very small burst of progeny (2.5 pfu per infected cell). A second burst of roughly the same size occurred 90 min later (Supplementary Figure 3a). An adsorption curve shows that ~74% of virions bound to cells in first 5 min, and >90% were bound by 20 min (Supplementary Figure 3b). Efficiency of lysogeny/mutation rate tests demonstrated 2 ± 1% of cells of the strain 919/174 are resistant to ΦcrAss001 on initial contact. Such a high incidence of resistant clones could be explained by either lysogenic/pseudolysogenic immunity, or resistance mediated by rapid events such as gene expression control or site-specific recombination. Thirty of these clones (potential lysogens) were initially PCR-positive for the presence of ΦcrAss001 (using the portal protein gene as a target) and were resistant to the bacteriophage in agar overlay plaque/spot assays. Successive rounds of propagation and re-purification from single colonies of 10 clones resulted in loss of the ΦcrAss001 PCR signal, but the phage-resistance phenotype was retained in all clones. Interestingly, when resistant clones were streaked out and 72 isolated colonies were picked, 2 of them demonstrated reversion to the phage-sensitive phenotype.

Infection of exponentially growing cells of *B. intestinalis* 919/174 with ΦcrAss001 at different MOIs did not result in complete culture lysis, but caused a delay in stationary phase onset time and final density at stationary phase (Supplementary Figure 3c). A brief lysis period occurred in the first few hours after infection, with timing dependent on the MOI, followed by recovery of bacterial growth. Interestingly, this could not be replicated in semi-solid agar cultures, where lysis was highly efficient and bacterial growth only occurred at MOI of <$10^{-2}$, which is in agreement with the ability of the bacteriophage to form substantial, clear plaques. In order to investigate long-term interactions between ΦcrAss001 and its host, bacterial cells infected at high MOIs were allowed to reach stationary phase and then passaged daily or bi-daily for a period of 23 days (Fig. 3e). Viral titre (pfu per mL), which on the first two passages reached ~$10^{10}$, was reduced and remained steady between $10^6$ and $10^8$ for the duration of the experiment. When isolated colonies were obtained from these cultures at the final time point, ~46% of clones demonstrated complete resistance (lack of spot formation in agar overlays), whereas the remainder of clones were either fully or weakly sensitive (hazy spots) to the ancestral bacteriophage.

and reached $10^{10}$ plaque-forming units (pfu) per mL when propagated in broth culture. Importantly, 14 other strains of six different *Bacteroides* species tested could not support growth of ΦcrAss001 in either liquid or semi-liquid media. Broad diversity

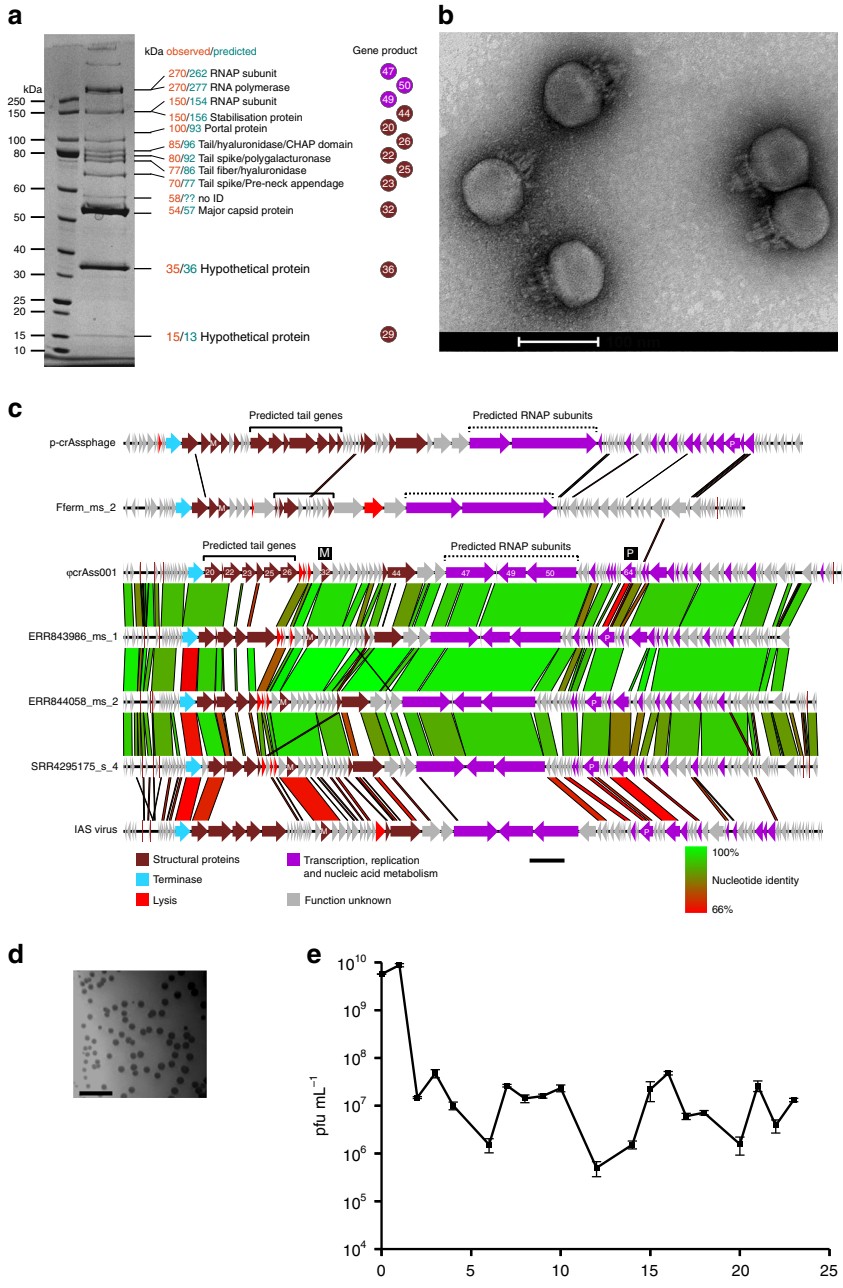

**Fig. 3** ΦcrAss001 virion structure, related bacteriophages and persistence in a continuous host co-culture. **a** SDS-PAGE analysis of protein content in ΦcrAss001 virions and identification of selected polypeptides using MALDI-TOF (see supplementary methods for details). **b** TEM image of uranyl acetate negatively contrasted ΦcrAss001 virions, scale bar is 100 nm (×62,000 magnification, accelerating voltage of 120 kV). **c** Genomic synteny and BLASTn nucleotide sequence homology between ΦcrAss001, a group of highly related uncultured bacteriophages (ERR843986_ms_1, ERR844058_ms_2, SRR4295175_s_4, see Guerin et al.[13] for details), and other members of the crAss-like virus family (p-crAssphage, IAS virus, bacteriophage Fferm_ms_2 propagated and visualised from a faecal fermentor system[13]); highly variable tail genes region is marked a solid bracket; region containing giant open reading frames of predicted RNA polymerase subunits is highlighted with dotted bracket; M, major capsid protein; P, DNA polymerase, scale bar is 5 kb. **d** Plaque morphology of ΦcrAss001 after 48 h incubation in a 0.3% FAA agar overlay with *B. intestinalis* 919/174 as host, scale bar is 1 cm. **e** Persistence of ΦcrAss001 in a continuous culture of bacteriophage infected *B. intestinalis* 919/174 with daily (or bi-daily) transfers for 23 days (pfu per mL, mean ± SD of three independent cultures incubated in parallel)

## Discussion

Collectively, our results suggest that ΦcrAss001 uses an unusual infection strategy to replicate in vitro on its *B. intestinalis* host very efficiently on semi-solid agar (giving rise to large clear plaques) and yet replication in liquid culture is done so without hampering proliferation of the host bacterium. Despite not being able to form true lysogens, the virus seems to be able to co-exist with its host in an equilibrium that is likely to confer an ecological advantage to both partners. Such long-term co-existence of virulent bacteriophages with their hosts in the mammalian gut without considerable reduction in titre of the latter has been demonstrated in multiple instances in germ-free, antibiotic pre-

treated and conventional models[23,24]. It is, however, unusual to observe the ability of presumably virulent bacteriophage to demonstrate similar behaviour in vitro. To a certain extent the phenomena we observed were comparable with unusual carrier state life cycle in *Campylobacter jejuni* viruses[25] or transient pseudolysogeny with virulent *Pseudomonas aeruginosa* viruses[26]. Taken together, these results are puzzling. Given that in liquid culture we used an MOI of 1 and observed a high adsorption rate, we cannot explain why two bursts seemed to occur and why the broth culture did not clear. We can conclude that the virus probably causes a successful lytic infection with a size of progeny per capita higher than 2.5 in a subset of infected cells (giving rise to a false overall burst size of ~2.5), and also enters an alternative interaction (pseudolysogeny, dormant, carrier state, etc.) with some or all of the remaining cells. Overall, this allows both bacteriophage and host to co-exist in a stable interaction over prolonged passages. The nature of this interaction warrants further investigation.

Longitudinal examinations of faecal viral communities in humans and germ-free mice with transplanted human viromes[13,27] support the hypothesis that crAssphages use an unusual strategy to establish themselves at high levels in the gut and to then persist stably within the microbial communities for several weeks to as long as several months. The absence of any detectable integrase gene and our inability to isolate stable lysogens, together with the lack of any evidence that this or any other crAss-like bacteriophages can occur in the form of prophages, suggest that pseudolysogeny and other mechanisms, such as physical binding of virions to the intestinal mucous gel[28] might be responsible for long-term persistence of crAss-like bacteriophages in vivo. It has been previously suggested that the mammalian gut environment, characterised by high microbial loads almost at the limit of carrying capacity, favours selection of temperate bacteriophages capable of peaceful long-term co-existence with their hosts[29] with moderate levels of genetic drift[30]. This seems to be true for other types of environments, with high microbial density negatively correlated with lower bacteriophage load and lysogeny favoured over lysis[31]. The ecological model explaining such phenomena is known as the Piggyback-the-Winner model[32], as opposed to the classical Kill-the-Winner hypothesis[22]. In this framework, it seems logical that crAss-like bacteriophages evolved to demonstrate lysogeny-like behaviour, resulting in stable colonisation of the gut by both the virus and its *Bacteroides* host in high numbers for prolonged periods of time. However, the reported ability of long-term persistence of crAss-like bacteriophages in vivo might be a result of multiple simultaneously acting factors, such as physical and physiological protection of host cells from infection, ability of bacteriophage to engage into unusual type of interaction with its host, rapid co-evolution of bacteriophage and host strain (Red Queen dynamics[22]), as well as other factors. Further studies will be required to fully understand the replication cycle of crAss-like bacteriophages, their peculiar ability to persist at high titres in the human gut microbiota, as well as their significance for human intestinal physiology and disease. This first report of a bacteriophage-host pair should accelerate our understanding of these highly abundant and unusual viruses.

## Methods

**Bacteriophage enrichment from human faecal samples.** Faeces were collected from 20 consenting Irish adult volunteers according to study protocol APC055, approved by the Cork Research Ethics Committee. Samples were collected (without fixative or preservative) in the volunteer's home and transported to the research facility at ambient temperature, avoiding exposure to heat and stored at −80 °C until processed.

Faecal filtrates were prepared by homogenising 0.5 g of thawed faeces in 10 mL SM buffer using vigorous vortexing for 5 min. Tubes were then chilled on ice for 5 min prior to centrifugation at $5200 \times g$ in a swing bucket rotor for 10 min at +4 °C.

Supernatants were transferred to new tubes and centrifugation was repeated once again. Supernatant were subsequently filtered twice through a 0.45 μm pore syringe-mounted polyethersulfone (PES) membrane filters. Pooled faecal filtrates were prepared by combining equal volumes of filtrates from 20 donors.

A total of 53 bacterial strains were used for the enrichment experiment (Supplementary Data 1). Strains were grown on YCFA broth[33] supplemented with a mixture of carbohydrates (D-glucose, soluble potato starch, D-cellobiose and D-maltose) at concentrations of $2\,\mathrm{g\,L^{-1}}$ of each. Hungate tubes were filled with 9 mL of pre-reduced YCFA broth in an anaerobic chamber and sterilised by autoclaving. Tubes were inoculated with 1 mL of overnight culture using 23 G hypodermic needles and incubated at 37 °C until the $OD_{600} = 0.3$ was achieved (CO8000 Cell Density Meter, Biochrom WPA). One millilitre of culture was then removed from the Hungate tube and replaced with the pooled faecal filtrate, maintaining a volume of 10 mL. Incubation was continued at 37 °C overnight. Cultures were centrifuged twice as described above and supernatants were filtered through 0.45 μm pore PES filters. One millilitre of this filtrate was then added to an exponentially growing culture of the same strain as described above. The enrichment process was repeated a total of three times.

**Extraction of VLP-associated DNA and shotgun sequencing.** The VLP DNA fraction was extracted from 10 mL bacterial cultures after the third round of phage enrichment. Briefly, cultures were centrifuged twice and filtered as described above. NaCl and PEG-8000 powders were then added to filtrates to give a final concentration of 0.5 M and 10% w/v, respectively. After complete dissolving, samples were incubated overnight (16 h) at +4 °C. The remaining DNA extraction steps were performed as described before[34]. Briefly, phage precipitates were collected by centrifugation at $5200 \times g$ for 20 min at +4 °C. Pellets were resuspended in 400 μL of SM buffer and extracted by gentle shaking with equal volume of chloroform followed by centrifugation at $2500 \times g$ for 5 min. The aqueous phase (~360 μl) was aspirated into clean Eppendorf tubes and treated with 8 U of TURBO DNase (Ambion/ThermoFisher Scientific) and 20 U of RNase I (ThermoFisher Scientific) in the presence of 1 mM $CaCl_2$ and 5 mM $MgCl_2$ at 37 °C for 1 h before inactivating enzymes at 70 °C for 10 min. This was followed with Proteinase K (40 μg) treatment in the presence of 0.5% SDS for 20 min at 56 °C. Viral particles were then lysed by addition of 100 μl of Phage Lysis Buffer (4.5 M guanidinium isothiocyanate, 44 mM sodium citrate pH 7.0, 0.88% sarkosyl and 0.72% 2-mercaptoethanol) at 65 °C for 10 min. Lysates were extracted twice with equal volume of phenol/chloroform/isoamyl alcohol 25:24:1 (Fisher Scientific) and subjected to final round of DNA purification using DNeasy Blood & Tissue Kit (Qiagen).

One microlitre of DNA sample was amplified using MDA technology with Illustra GenomiPhi V2 kit (GE Healthcare). The latter step was done in triplicate for each sample. Products from all three MDA reactions were pooled together and subjected to additional round of purification using DNeasy Blood & Tissue Kit. Amplified DNA was quantified using Qubit dsDNA HS Assay Kit (Invitrogen/ThermoFisher Scientific) and subjected to random shotgun library preparation using Nextera XT DNA Library Preparation Kit (Illumina, cat #FC-131-1096) and bead-based normalisation following the standard manufacturer's protocol. Ready-to-load libraries were sequenced using a proprietary modified protocol using $2 \times 300$ bp paired-end chemistry on an Illumina HiSeq 2500 platform at GATC Biotech AG, Germany.

In order to confirm the genome sequence of ΦcrAss001 and investigate the structure of bacteriophage genome termini and packaging mode an additional sequencing effort was made using Illumina Hiseq platform on the plaque-purified bacteriophage DNA. Briefly, bacteriophage was propagated on 10 mL broth culture of *B. intestinalis* 919/174 as described below. Phage particles were precipitated from the filtrate with NaCl/PEG-8000 and DNA was extracted as described before[34] except that the phenol/chloroform extraction step was omitted. DNA was then quantified using Qubit dsDNA HS Assay Kit and subjected to shotgun library preparation using TruSeq Nano DNA kit (Illumina, cat #20015965) after random DNA shearing by sonication as recommended in the kit manual. Library was sequenced using Illumina HiSeq 2500 at GATC Biotech AG, Germany.

**Analysis of shotgun sequencing data and genome annotation.** Raw Illumina paired-end reads were quality-checked using FastQC v0.11.5, trimmed and filtered using Trimmomatic v0.36[35] using sliding window approach with window size of 4 nt and minimum allowed Phred score of 20. In addition, all reads were cropped to a length of 230 nt with first 10 nt removed. Reads shorter than 60 nt were discarded. Filtered reads were assembled on per-sample basis using metaSPAdes v3.10.0[36] with standard parameters. Contigs shorter than 1000 nt were discarded. Contigs from all enrichment samples were then pooled together and demultiplexed by picking the longest representatives for each group of contigs with >90% sequence identity and >90% of sequence overlap (as determined by BLASTn v2.2.28+[37]). To quantify presence of various contigs in the enrichment samples, filtered reads were aligned back to the common demultiplexed database of contigs using Bowtie v2.1.0[38] in the end-to-end mode. Counts of aligned reads were extracted from alignment data using Samtools v0.1.19.

Annotation of genomic contig representing phage ΦcrAss001 was done using VIGA v0.10.3[18] with BLASTp searches against NCBI nr database (snapshot of 2018-01-15), and HMM searches against UniProt/Swiss-Prot database (snapshot of 2018-01-24). Additionally, amino-acid sequences of the encoded proteins were

annotated using HHpred Web-server[19] [https://toolkit.tuebingen.mpg.de/#/tools/hhpred] with HMM profile-profile searches against the following databases: PDB mm_CIF70_25_Feb, Pfam-A v31.0, NCBI CD v3.16 and TIGRFAMs v15.0. Circular map of φcrAss001 was visualised using GView v1.7. Nucleotide sequence homology with related bacteriophages was visualised using Easyfig v. 2.2.2 with the following parameters: search algorithm—BLASTn, e-value cut-off 0.001, length filter 0. Analysis of bacteriophage genome termini and mode of packaging was performed using PhageTerm v1.0.10[17].

For analysis of CRISPR spacer and tRNA content in the ΦcrAss001 genome the following approach was used. A collection of 5591 bacterial genomes (both complete and draft assemblies) were obtained from NCBI RefSeq database, release 89. CRISPR repeats and tRNA genes were predicted using PILER-CR v1.06 (default settings) and ARAGORN v1.2.36[39] (with -fon flag), respectively. BLASTn search of CRISPR spacers against the ΦcrAss001 was performed using -task "blastn-short" parameter. BLASTn search of tRNA genes was performed similarly, but with default parameters. None of the predicted tRNA genes had significant matches against the phage genome. A total of 78,371 CRISPR spacer hits were obtained, which were then sorted by Bitscore and 3000 best candidates were plotted in Supplementary Figure 2.

The PCR primers ΦcrAss001-F1 (5′-AATAAGGTGGAAGATGCTGAC-3′) and ΦcrAss001-R1 (5′-TTATCCATTTGGTCAACAGCTC-3′), specific towards gene 20 (portal protein) of ΦcrAss001 were developed for detection and quantification of the phage in cultures of B. intestinalis 919/174.

**Homology modelling of ΦcrAss001 MCP.** Homology modelling of MCP protein tertiary structure was performed using HHPred-MODELLER online suit[20]. Initial HMM-HMM search was performed against the PDB_mmCIF70_28_Jul database. Staphylococcus phage 812 was selected as the template for modelling. PIR format alignments were submitted to MODELLER and the generated.pdb files were rendered using CLC Main Workbench 8.0.

**Propagation and biological properties of ΦcrAss001.** The host strain B. intestinalis 919/174 was routinely maintained in Fastidious Anaerobe Broth (FAB, Oxoid) anaerobically at 37 °C. Cultures were infected with ΦcrAss001 at various MOIs (MOI = 1 for optimal phage yield) in early logarithmic phase of growth ($OD_{600} = 0.2$, corresponding to $\sim 2 \times 10^8$ colony-forming units per mL) with or without addition of $CaCl_2$ and $MgCl_2$ to final concentrations of 1 mM each. Infected cultures were collected after overnight incubation, centrifuged at $5200 \times g$, 4 °C for 15 min to remove cells and then filtered through 0.45 μm pore PES syringe-mounted membrane filters. Phage cultures could be stored at +4 °C without any further treatment for a period of up to 3 months without significant loss of titre. Plaque and spot assays were performed in a standard manner using 3 mL of 0.4% Bacto agar (Becton Dickinson) for overlays on 100 mm diameter plates with fastidious anaerobic agar (FAA) solid agar. Two hundred microlitres of overnight B. intestinalis 919/174 culture in FAB and 100 μL of phage sample were added to molten overlay agar tubes kept at 45 °C, followed by vortexing and pouring on pre-made FAA plates. Plates were incubated anaerobically at 37 °C for 24 h before plaque counting.

For the one-step growth experiment, early logarithmic phase culture of B. intestinalis 919/174 was infected at an MOI of 0.7 for 5 min at room temperature, followed by centrifugation at $5200 \times g$, 4 °C for 10 min, removal of supernatant and re-suspending of the infected cells in fresh FAB medium. Incubation was continued anaerobically at 37 °C for further 225 min with removal of 1 mL samples every 15 min. Samples were filtered through 0.45 μm pore PES filters and subjected to standard plaque assays with appropriate dilutions.

Adsorption experiments were performed in a similar manner. Early logarithmic phase culture B. intestinalis 919/174 were mixed with ΦcrAss001 at an MOI of 1 and incubated at room temperature for 1 h with removal of aliquots every 5 min followed by immediate filtering through 0.45 μm pore PES filters and subsequent standard plaque assay.

Efficiency of lysogeny experiments were conducted by spreading 200 μL of phage lysates ($10^9 - 10^{10}$ pfu per mL) onto 100 mm diameter FAA agar plates. After fully dried, both phage-covered and negative control plates were inoculated with serial 10-fold dilutions of B. intestinalis 919/174 overnight culture. Efficiency of lysogeny was determined as a percentage of colonies observed on phage-coated plates relative to the total counts on negative control plates after 48 h of anaerobic incubation at 37 °C. Thirty clones were randomly picked from plates, streaked out and subjected to PCR with ΦcrAss001 portal protein gene-specific primers. All were found positive and resistant to phage in agar overlay spot assays. After three additional rounds of streaking with isolation of single isolated colonies 10/10 clones were PCR-negative but still resistant to ΦcrAss001.

The long-term phage/host co-cultivation experiment was performed as follows. Ten millilitre liquid cultures (n = 3) of B. intestinalis 919/174 were infected at $OD_{600} = 0.2$ at MOI = 1. Incubation was continued overnight. The obtained phage/host mix was sub-cultured the next day into fresh FAB medium at 1:50 ratio. Sub-culturing was repeated daily or bi-daily, 17 times for the period of 23 days. Bacterial culture demonstrated normal growth throughout the experiment and reached typical $OD_{600}$ 0.74–1.00 at stationary phase. Phage proliferation was monitored by performing standard plaque assays at each sub-culture (n = 19,

Fig. 3e). At the end of experiment cultures were streaked out on plates and 24 colonies were randomly picked (8 per plate). Clones were sub-cultured twice and subjected to PCR with ΦcrAss001 portal protein gene-specific primers. All colonies were found negative. In agar overlay spot assays 11/24 clones were completely resistant to phage, while the remaining 13/24 were either fully or partially (hazy spots) sensitive.

**TEM and proteomic analysis of ΦcrAss001.** Sixty millilitres of phage lysate obtained as described above was ultra-centrifuged at $120,000 \times g$ for 3 h using a F65L-6x13.5 rotor (ThermoScientific). The resulting pellets were resuspended in 5 mL SM buffer. The viral suspensions were further purified by overlaying them onto a caesium chloride (CsCl) step gradient of 5 and 3 M solutions, followed by centrifugation at $105,000 \times g$ for 2.5 h. A band of viral particles visible under side illumination was collected and buffer-exchanged using three sequential rounds of 10-fold diluting and concentrating to the original volume by ultra-filtration using Amicon Centifugal Filter Units 10,000 molecular weight cut-off (MWCO; Merck). Following this, 5 μL aliquots of the viral fraction were applied to Formvar/Carbon 200 Mesh, Cu grids (Electron Microscopy Sciences) with subsequent removal of excess sample by blotting. Grids were then negatively contrasted with 0.5% (w/v) uranyl acetate and examined at UCD Conway Imaging Core Facility (University College Dublin, Dublin, Ireland) by Tecnai G2 12 BioTWIN transmission electron microscope. The same CsCl gradient fraction was further concentrated 10-fold using Amicon Ultra-0.5 Centrifugal Filter Unit with 3 kDa MWCO membrane (Merck, Ireland). The obtained sample (25 μL) was loaded onto a pre-made Bolt 4-12% Bis-Tris Plus reducing SDS-PAGE gel (Invitrogen) and separated at 200 V for 30 min using 1× NuPAGE MOPS SDS Running Buffer. Thirteen detectable bands with approximate molecular weights of 15, 35, 54, 58, 70, 77, 80, 85, 100, 150, 270 and >300 kDa were excised and subjected to MALDI-TOF/TOF (Bruker Ultraflex III) protein identification following in-gel trypsinization, at Metabolomics & Proteomics Technology Facility (University of York, York, UK).

**Reporting Summary**. Further information on research design is available in the Nature Research Reporting Summary linked to this article.

## Data availability
Data that support these findings have been deposited into NCBI databases with the following accession codes: Bioproject PRJNA486819 (bacteriophage crAss001 genomic DNA, raw reads); GenBank MH675552 (bacteriophage crAss001 assembled and annotated genome); GenBank QRES00000000 (bacteriophage host B. intestinalis APC919/174 assembled and annotated genome). Source data for Figs. 2 and 3, as well as for Supplementary Figures 1, 2 and 3 are provided as a Source Data file.

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

## Acknowledgements

Authors thank Dr Dimitri Scholz and Tiina O'Neill, UCD Conway Institute of Biomolecular and Biomedical Research, Dublin, Ireland, for their help with transmission electron microscopy imaging. This research was conducted with the financial support of Science Foundation Ireland (SFI) under Grant Number SFI/12/RC/2273, a Science Foundation Ireland's Spokes Programme, which is co-funded under the European Regional Development Fund under Grant Number SFI/14/SP APC/B3032, and a research grant from Janssen Biotech, Inc.

## Author contributions

A.N.S., E.V.K. and C.B.F. performed the experiments; A.N.S., S.R.S., L.A.D., R.P.R. and C.H. analysed the data; R.P.R. and C.H. supervised the project.

## Additional information

**Competing interests:** The authors declare no competing interests.

