## [Peer Review File · Nature Communications]

Reviewers' comments:

Reviewer #1 (Remarks to the Author):

The Short Note from A. Shkoporov and colleagues from UC Cork is of interest since it represents the first in vitro cultivation of an crAss phage, which are abundant and ubiquitous gut phages, but so far only known as in silico phages from metagenome projects. Its high titer in so many human stool samples suggests that it displays properties of successful adaptation to growth in the gut ecosystem. Possessing an isolate and an in vitro propagation system for this phage opens now new possibilities of investigating properties needed for efficient phage propagation in the gut. Indeed a number of interesting observations were reported. Despite this overall positive appreciation of this contribution, a number of clarification would substantially increase the value of the this contribution to readers of Nature Communication.

I. 65-66: The 8 % difference between the GC content of the crAss phage and *B. intestinalis*, the propagating host, suggests that the phage has not evolved on this bacterial host. Please comment on the possibility of a much larger host range of phages in their natural environment (e.g. the recent Polz Nature paper), the low burst size seems to support this suspicion and there is a substantial eco-evo literature on this subject. Are CRISPR sequences of crAss phages in the *B. intestinalis* genome?

I. 74-75: crAss phages characterized by Yutin et al. show the gene constellation terminase-portal-major head gene, well known from Siphoviruses of many evolutionary lineages. Here the major capsid gene is relatively far away- apparently tail genes were transposed into the capsid module. Additional alignments with other Assvirinae than IAS in Fig. 2C would be helpful to see to what extent synteny is conserved in this phage group. Can it be deduced whether crAss001 phage shows the HK97 or lambda head fold?

I. 80-82: The phage is unusual in having a DNA polymerase and even more exotic by possessing an RNA polymerase with two additional subunits transcribed from adjacent, but opposite strands. This would create difficulties in constituting a functional RNA polymerase. Could this be the reason that the RNA polymerase comes as a structural protein in the virion? Otherwise, this constellation with subunits transcribed from opposite strands would open interesting regulatory possibilities. In view of the interest, confirmation of this genomic constellation by diagnostic PCR would be helpful. What is the constellation in other crAss phages?

I. 82-84: the region between genes 99 and 8 looks like a non-phage mobile DNA element flanked by two HNH endonucleases that has invaded the phage genome. This phage is also target of further invasion by mobile DNA elements as seen by isolated group I intron endonuclease and HNH endonuclease genes elsewhere in the genome. The situation reminds coliphage T4, whose natural habitat is also the gut. Reference to this literature might orient the reader. What is the situation for these mobile genes in the in silico characterized crAss phages?

I. 89-94: The tail structure in Fig. 2B is substantially different from the crAss phage obtained from the fecal fermenter system of Fig. 7 from ref. 11, the latter are T7 tail spike-like, while the former show a much more elaborated structure with a complicated baseplate structure. A genome comparison with the fermenter phage could be revealing to localize tentatively the additional genes encoding the baseplate genes.

I. 95-96: please define where this isolate belongs in the alpha to delta Assvirinae classification scheme.

I. 97-101: possible duplication /overlap with Ref. 11.

I. 114-118: This is not a reliable lysogeny test as also seen from the variable PCR results, but it probably identified cells that were physiologically insensitive to phage infection. Distinction of genetically identical, but physiologically distinct cells (e.g. by distinct transcription pattern) that differ for phage sensitivity might be interesting future research activities. The authors come back to this point in the final paragraph discussing lysogeny, pseudolysogeny as possible causes for the peculiar phage-bacterium interaction. However, similar observations have been made with conventional virulent coliphages (T7 and T4) infecting *E. coli* in monocolonized mice (Weiss Virology 2009; Maura Env Micro 2012), pointing to physiological differentiation of bacterial cells that confer non-genetic phage resistance achieving in the gut long term parallel persistence of

phage and target cells. Since crAss phage shows this property in vitro (the low burst size, the multiple phage peaks would point into that direction), this is a fertile future research direction which should at least be mentioned in this report.

L. 129: I am not comfortable with this statement: filamentous coliphages leave the cell without causing lysis, but this needs a special exit mechanism for cell membrane transfer without disrupting its integrity. However, this phage has lysins and holins, which makes this process unlikely. Consider alternative interpretations, e.g. those mentioned in the previous paragraph.

L. 148-149: This is unlikely since the mucus in the gut is physiologically a conveyor belt, which would lead to rapid phage elimination.

Harald Brüssow

Reviewer #2 (Remarks to the Author):

This short communication is well written and presented. It provides the first identification of the bacterial host of one of the member of the crAssphage family, which is the most abundant phage family residing in human gut. Since 4 years, the international phage community was puzzled by the difficulties to find the host for such an abundant phage. The authors brilliantly succeeded in this task and provide the first biological description of the life cycle of this phage.

Comments/observations are:

They are few "phages" instead of "bacteriophages" in the manuscript (ex L123,L125). Please correct.

L1: I would add "intestinalis", as Bacteroides was already suggested as a host for crAssphage.

L79: authors mentioned genes involved in recombination but in L85 they mentioned that no identifiable recombinase genes were identified. Please clarify

L117: please indicate the number of clones on which these tests (loss of positive PCR) were performed.

L123: Rephrase "fate of phage in co-culture" by "long-term interactions between crAss001 and its host, bacterial cells infected at high MOI..." for example. Please add a detailed section of this procedure in the supplement as I couldn't find it. Did you look at some isolated cells over time to determine if they were individually sensitive or resistant?

Here authors could expand the discussion by referring to the following literature reporting similar observation from the feces of mice exposed to coliphages (PMID: 22118225, 29174401, 29914064).

L130: why do you exclude pseudolysogeny here (if so how did you test it?) and later L147 consider it as plausible explanation? About pseudolysogeny authors may look at this paper: PMID 26921273.

L159: I cannot see the Orange color: it looks like tRNA are colored in black. Also there is a think line starting from gene 48 and going to gene 50 counterCW that should be removed or explained?

L165: Could the authors comment on the fact that the observed Mass is always different to the predicted. Is mass spec so imprecise or is it a sign of post-translational modification?

L170: the reference for Guerin is 12, not 1.

In figure 1: insert the name of crAss001 in the center of the circle as well as the total length.

In Figure 2: panel C, replace the asterisk by a horizontal line as this variable region is not localized into a single ORF, correct?

Panel C: is the original crAssphage from Dutilh amongst the three chosen, and if not why not including it?

In the entire supplement: it looks like the crAss001 carried another name before: APC-LOC110. Please update

Figure S2C: the legend is not precise enough to determine the difference between these 4 panels and the color code for the MOI is not appropriate: what is the color of the control with no phage? With only 5 curves you can chose 5 different colors without using a graded scale. How do you define "sloppy agar"

L51 add "for each of the 53 samples" after library preparation.

L112: ass the protocol for Fig 2E

Overall, since the phage was enriched from human samples, did the authors try to plaque purify this phage from the fecal material using an overlay of the strain of *B. intestinalis* they used to enrich t? If this is unsuccessful it would suggest that the phage that was enriched has developed/acquired/mutated some characteristic to amplify in broth. Would it then be possible to match the reads from the viral fecal filtrate of the original fecal samples on the purified crAss001 to 1) show that the original sample contain this phage and 2) eventually identify variants.

Reviewers' comments:

Reviewer #1 (Remarks to the Author):

The Short Note from A. Shkoporov and colleagues from UC Cork is of interest since it represents the first in vitro cultivation of an crAss phage, which are abundant and ubiquitous gut phages, but so far only known as in silico phages from metagenome projects. Its high titer in so many human stool samples suggests that it displays properties of successful adaptation to growth in the gut ecosystem. Possessing an isolate and an in vitro propagation system for this phage opens new possibilities of investigating properties needed for efficient phage propagation in the gut. Indeed a number of interesting observations were reported. Despite this overall positive appreciation of this contribution, a number of clarification would substantially increase the value of the this contribution to readers of Nature Communication.

Response: We are grateful to Dr. Harald Brüssow (Reviewer 1) for his overall positive assessment of our study and for his very thoughtful and detailed critical comments. We tried our best to address as much of Dr. Brüssow's comments as possible and to answer all of his questions.

I. 65-66: The 8 % difference between the GC content of the crAss phage and *B. intestinalis*, the propagating host, suggests that the phage has not evolved on this bacterial host. Please comment on the possibility of a much larger host range of phages in their natural environment (e.g. the recent Polz Nature paper), the low burst size seems to support this suspicion and there is a substantial eco-evo literature on this subject.

Response: Analysis of the recently obtained draft genome sequence of the actual host *B. intestinalis* APC919/174 (GenBank ID MH675552) shows GC mol% of 42.5% (which is indeed 7.8% different from the bacteriophage). We added the notion that this may suggest that crAss001 did not evolve on this particular host, and/or it may have other hosts in the environment. We added a brief discussion of this fact in lines 140-155. We also attempted to get additional insights into potential additional hosts or former hosts of this bacteriophage (in case host switching had recently occurred) using CRISPR spacer matching approach. The results were inconclusive and presented in the new Fig. S2. At the same time we can not fully agree with the reviewer that substantial difference in the G+C content in natural bacteriophage-host pairs is entirely uncommon. One classical example can be bacteriophage T4 (35.5 mol%) and *E. coli* (45.5-51.1 mol%).

Our host range tests revealed that the bacteriophage crAss001 was unable to replicate on a different strain of *B. intestinalis* (DSM 17393), as well as other members of genus *Bacteroides*, including *B. caccae* (2 strains), *B. cellulosilyticus* (1 strain), *B. ovatus* (2 strains), *B. uniformis* (2 strains), *B. vulgatus* (3 strains, see Table S1 for details). Our recent preprint article (Guerin et al., Ref 11) revealed a very high diversity of crAss-like bacteriophages in the human gut, 244 individual strains identified in metagenomic data, belonging to 10 genus- and 4 subfamily-level candidate taxa. We have also observed a case of stable colonization for a 1-year period of a single human individual with up to 7 strains (6 genera) per sample representing up to 30% of faecal virome. We believe that these direct and indirect forms of evidence, together with previous unsuccessful attempts to isolate crAss-like bacteriophages in culture (Ref 8), suggest that crAss-like bacteriophages are narrow specialists, rather than generalists in terms of their host range.

We also tried to discuss our results in the context of recent bacteriophage evolutionary ecology concepts (lines 190-196 and 213-226).

Are CRISPR sequences of crAss phages in the *B. intestinalis* genome?

Response: Our preliminary analysis of *B. intestinalis* APC919/174 genome showed it has no detectable CRISPR arrays and no Cas genes. A lack of a CRISPR loci in other 3 *B. intestinalis* strains included into NCBI RefSeq database prevents us from using CRISPR arrays to determine whether or not these strains may also act as a host for crAss001 or other crAssphages.

l. 74-75: crAss phages characterized by Yutin et al. show the gene constellation terminase-portal-major head gene, well known from Siphoviruses of many evolutionary lineages. Here the major capsid gene is relatively far away- apparently tail genes were transposed into the capsid module. Additional alignments with other Assvirinae than IAS in Fig. 2C would be helpful to see to what extent synteny is conserved in this phage group. Can it be deduced whether crAss001 phage shows the HK97 or lambda head fold?

Response: To highlight differences in the gene order we added two additional genomes to Fig. 2C: prototypical crAssphage (p-crAssphage, candidate genus I, *Alphacrassvirinae* according to classification proposed in Ref 11) and a crAss-like phage contig Fferm_ms_2 which dominated a sample used in faecal fermentation in the same study (Ref 13). This latter contig belongs to the candidate genus V (subfamily *Gammacrassvirinae*). As it is evident from updated Fig. 2C, crAss001 and related phages (candidate genus VI, subfamily *Betacrassvirinae*) show a somewhat different type of genomic organization compared to other crAss-like phages. Major capsid protein gene is transposed behind the tail gene module, one of the giant ORFs, comprising putative RNA-polymerase is split in two oppositely oriented ORFs, DNA polymerase gene is positioned differently.

As far as we know, bacteriophage lambda capsid has the same type of fold as HK97 ([https://www.cell.com/fulltext/S0969-2126\(08\)00292-X](https://www.cell.com/fulltext/S0969-2126(08)00292-X)). Maybe the reviewer means DJR fold? We performed homology modelling of crAss001 MCP along with p-crAssphage and IAS virus MCPs and found out they had a classical HK97-type fold, as was expected. Please see lines 97-102 and the new Fig. 2.

l. 80-82: The phage is unusual in having a DNA polymerase and even more exotic by possessing an RNA polymerase with two additional subunits transcribed from adjacent, but opposite strands. This would create difficulties in constituting a functional RNA polymerase. Could this be the reason that the RNA polymerase comes as a structural protein in the virion? Otherwise, this constellation with subunits transcribed from opposite strands would open interesting regulatory possibilities. In view of the interest, confirmation of this genomic constellation by diagnostic PCR would be helpful. What is the constellation in other crAss phages?

Response: We appreciate the positive comments about the novelty; we have added a short statement to reflect this comment and potential regulatory activity in the manuscript (lines 133-137). As it is demonstrated in Fig 3c, as well as elsewhere (Yutin et al., 2018) crAss001, IAS virus and other representatives of candidate genus VI have similar arrangement of putative RNA-polymerase subunit genes. They were sequenced with different chemistries (454, Illumina), assemblies on them were done using different software and by different groups, so it is unlikely, that we are facing a case of miss-assembly here. We report additional sequencing using different chemistry at a higher relative coverage (please see Supplementary methods, sections **Extraction of VLP-associated DNA and shotgun sequencing, Analysis of shotgun sequencing data and annotation of pcrAss001 genome**) which yielded the same assembly of this region. Use of alternative assemblers (IDBA-UD, Megahit), as well as alignment of individual reads to the region in question using Bowtie2 also confirmed that (data not presented in the manuscript).

l. 82-84: the region between genes 99 and 8 looks like a non-phage mobile DNA element flanked by two HNH endonucleases that has invaded the phage genome. This phage is also target of further invasion by mobile DNA elements as seen by isolated group I intron endonuclease and HNH

endonuclease genes elsewhere in the genome. The situation reminds coliphage T4, whose natural habitat is also the gut. Reference to this literature might orient the reader. What is the situation for these mobile genes in the in silico characterized crAss phages?

Response: We appreciate the comment, but we do not necessarily agree that this region, which the reviewer probably defined based on it being flanked by two HNH homing endonuclease genes (intron endonucleases), is indeed non-bacteriophage DNA. First of all, the region, despite appearing contiguous on the circular genetic map, in fact corresponds to the bacteriophage termini when the genome linearizes itself. It includes most of the tRNA genes present in the bacteriophage genome as well as at least two proteins with potential role in protection against oxidative stress (DpsA and thioredoxin-ferredoxin reductase). It is not clear why this region could be the target for self-splicing intron integration, but several introns were also observed in various locations in prototypical crAssphage and IAS virus (Yutin et al., 2018).

l. 89-94: The tail structure in Fig. 2B is substantially different from the crAss phage obtained from the fecal fermenter system of Fig. 7 from ref. 11, the latter are T7 tail spike-like, while the former show a much more elaborated structure with a complicated baseplate structure. A genome comparison with the fermenter phage could be revealing to localize tentatively the additional genes encoding the baseplate genes.

Response: We appreciate this comment and agree that this is a very interesting observation. The most common type of virion in that study presumably corresponds to the most abundant sequence detected by metagenomic sequencing of the sample without MDA amplification, phage contig Fferm_ms_6 (candidate genus V). The high level of sequence divergence in the crAssphage family makes it virtually impossible to draw one-to-one correspondence between structural genes. We can rely more on the order of genes than on sequence comparison. It is now highlighted on the modified Fig. 2c, to which Fferm_ms_6 and p-crAssphage were added. There is a considerable difference in how the tail gene clusters are organized in candidate genera I, V and VI. Potentially, fewer identifiable tail genes in Fferm_ms_6 could reflect a simpler tail structure. This is now discussed in lines 117-121.

l. 95-96: please define where this isolate belongs in the alpha to delta Assvirinae classification scheme.

It's Candidate Genus VI, subfamily *Betacrassvirinae*. (see lines 123-124)

l. 97-101: possible duplication /overlap with Ref. 11.

Response: Not exactly complete overlap. Here we looked at relative abundance of this exact bacteriophage vs. relative abundance of candidate genus VI as a whole. Only the latter part has been presented in Guerin et al. paper.

l. 114-118: This is not a reliable lysogeny test as also seen from the variable PCR results, but it probably identified cells that were physiologically insensitive to phage infection. Distinction of genetically identical, but physiologically distinct cells (e.g. by distinct transcription pattern) that differ for phage sensitivity might be interesting future research activities. The authors come back to this point in the final paragraph discussing lysogeny, pseudolysogeny as possible causes for the peculiar phage-bacterium interaction. However, similar observations have been made with conventional virulent coliphages (T7 and T4) infecting *E. coli* in monocolonized mice (Weiss Virology 2009; Maura Env Micro 2012), pointing to physiological differentiation of bacterial cells that confer non-genetic phage resistance achieving in the gut long term parallel persistence of phage and target cells. Since crAss phage shows this property in vitro (the low burst size, the multiple

phage peaks would point into that direction), this is a fertile future research direction which should at least be mentioned in this report.

Response: We totally agree with this point raised by Dr. Brüssow and we thank him for directing us towards the very relevant examples in the literature. There are a number of possibilities here and we briefly discuss them (within the limited space of a short communication) in lines 161-163, 190-196. We agree that this is indeed a very interesting problem and a fertile future research direction.

The low burst size phenomenon, whatever the mechanistic explanation for that, seems to be important in supporting continuous replication and persistence of the phage in its natural habitat without significant reduction of the population of its bacterial host. We already mentioned a case of long persistence of up to seven crAss-like phage in a human host. Other evidence comes from an earlier study (Reyes et al., 2013) where a crAss-like human phage ϕ HSC05 (albeit not recognized as crAss-like at that time) were capable of stable colonization and long term persistence in mice. In our opinion there could be two mechanistic explanations for the low burst size: 1) each infected cell produces only ~ 2.5 new phage particles upon lysis; 2) some infected cells produce >2.5 new phage (10, 20, 50?) particles, while in others lytic replication is inhibited or delayed, no lysis follows, which results in an average per capita output of ~ 2.5 . The latter explanation better suits with our observations that crAss001 is unable to clear liquid host culture even when added at relatively high MOI (Fig. S3c). Elucidation of this mechanism will be the focus our future research.

l. 129: I am not comfortable with this statement: filamentous coliphages leave the cell without causing lysis, but this needs a special exit mechanism for cell membrane transfer without disrupting its integrity. However, this phage has lysins and holins, which makes this process unlikely. Consider alternative interpretations, e.g. those mentioned in the previous paragraph.

Response: I believe the disputed claim is: “ ϕ crAss001 uses an unusual infection strategy to replicate *in vitro* on its *B. intestinalis* host very efficiently... in liquid culture *without causing lysis* of the host bacterium.” Corrected to “replicate in liquid culture *without hampering proliferation* of the host bacterium”

l. 148-149: This is unlikely since the mucus in the gut is physiologically a conveyor belt, which would lead to rapid phage elimination.

Response: We appreciate this comment and mostly agree with it. There are however some studies demonstrating that binding to mucous gel can sequester the phage and slow down its elimination. For instance, sequestration of phage *in vivo* in the absence of its host was demonstrated in one of Dr. Brüssow’s studies (Weiss et al., 2009) where phage T7 administered to germ-free animals could re-grow one week after, when its host administration followed.

Reviewer #2 (Remarks to the Author):

This short communication is well written and presented. It provides the first identification of the bacterial host of one of the member of the crAssphage family, which is the most abundant phage family residing in human gut. Since 4 years, the international phage community was puzzled by the difficulties to find the host for such an abundant phage. The authors brilliantly succeeded in this task and provide the first biological description of the life cycle of this phage.

Response: We are grateful to Reviewer 2 for very positive feedback and thoughtful comments.

Comments/observations are:

They are few “phages” instead of “bacteriophages” in the manuscript (ex L123,L125). Please correct.

Response: Corrected throughout to either “bacteriophage” or “virus”.

L1: I would add “intestinalis”, as Bacteroides was already suggested as a host for crAssphage.
L79: authors mentioned genes involved in recombination but in L85 they mentioned that no identifiable recombinase genes were identified. Please clarify

Response: Added species name and corrected “integrase and recombinase” to just “integrase”.

L117: please indicate the number of clones on which these tests (loss of positive PCR) were performed.

Response: Originally 30 colonies were picked and found to be PCR positive after 1 re-streaking. Additional 3 cycles of re-streaking and propagation were performed with 10 clones. Please see corrections in lines 163-169.

L123: Rephrase “fate of phage in co-culture” by “long-term interactions between crAss001 and its host, bacterial cells infected at high MOI...” for example. Please add a detailed section of this procedure in the supplement as I couldn’t find it.

Response: Corrected.

Did you look at some isolated cells over time to determine if they were individually sensitive or resistant?

Response: Cultures were streaked out and ~50% of isolated colonies were found to be either completely resistant (lack of spot formation in agar overlays) or poorly sensitive (very hazy, almost invisible spots), whereas the remainder of clones were found sensitive to the bacteriophage (lines 180-183).

Here authors could expand the discussion by referring to the following literature reporting similar observation from the feces of mice exposed to coliphages (PMID: 22118225, 29174401, 29914064).

Response: Refs 23 and 24 were added at line 193.

L130: why do you exclude pseudolysogeny here (if so how did you test it?) and later L147 consider it as plausible explanation? About pseudolysogeny authors may look at this paper: PMID 26921273.

Response: We thank the reviewer for this comment. We think that transient pseudolysogeny indeed could be a plausible explanation for some of the observed phenomena. A thorough investigation of these possibilities would be the basis of a completely separate study. We clarified that in the text and added suggested Ref 26 at line 196.

L159: I cannot see the Orange color: it looks like tRNA are colored in black. Also there is a think line starting from gene 48 and going to gene 50 counterCW that should be removed or explained?

Response: The colour of tRNA genes was fixed. That line was absent in our original images. We believe that was just a technical error when building PDF files.

L165: Could the authors comment on the fact that the observed Mass is always different to the predicted. Is mass spec so imprecise or is it a sign of post-translational modification?

Response: The size of polypeptides was determined using SDS-PAGE, not MALDI-TOF (such large polypeptides cannot be resolved using MALDI-TOF). MALDI-TOF was used for analysis of molecular masses of peptides resultant from trypsin degradation of the proteins extracted from gel slices. The composition of peptide fragments released from each slice (peptide fingerprint) was then compared to peptide profiles predicted based on known amino acid sequences. There could be multiple reasons for moderate deviation of observed molecular weights of proteins in SDS-PAGE from calculated ones: imprecision of SDS-PAGE, presence of post-translational modifications or anomalous electrophoretic mobility of the polypeptides due to peculiar amino acid content (see added lines 109-111). None of them are rare or uncommon in practice when SDS-PAGE is used.

L170: the reference for Guerin is 12, not 1.

Response: Corrected.

In figure 1: insert the name of crAss001 in the center of the circle as well as the total length.

Response: Corrected.

In Figure 2: panel C, replace the asterisk by a horizontal line as this variable region is not localized into a single ORF, correct?

Response: Corrected.

Panel C: is the original crAssphage from Dutilh amongst the three chosen, and if not why not including it?

Response: The idea of this image was to compare crAss001 with some of the most related crAss-like phages from candidate genus VI. We now modified this image to include p-crAssphage and phage Fferm_ms_2 which dominated a sample used in faecal fermentation in Guerin et al., 2018 study (Ref 13) and supposedly was captured on TEM images there.

In the entire supplement: it looks like the crAss001 carried another name before: APC-LOC110. Please update

Response: Corrected.

Figure S2C: the legend is not precise enough to determine the difference between these 4 panels and the color code for the MOI is not appropriate: what is the color of the control with no phage? With only 5 curves you can choose 5 different colors without using a graded scale. How do you define “sloppy agar”

Response: Corrected.

L51 add “for each of the 53 samples” after library preparation.

Response: Corrected (line 65).

L112: add the protocol for Fig 2E

Response: Corrected.

Overall, since the phage was enriched from human samples, did the authors try to plaque purify this phage from the fecal material using an overlay of the strain of *B. intestinalis* they used to enrich it? If this is unsuccessful it would suggest that the phage that was enriched has developed/acquired/mutated some characteristic to amplify in broth. Would it then be possible to match the reads from the viral fecal filtrate of the original fecal samples on the purified crAss001 to 1) show that the original sample contain this phage and 2) eventually identify variants.

Response: We were not able to obtain plaques directly from faecal filtrates. However, sequencing of the pooled faecal sample filtrate yielded 901 reads aligning to the crAss001 genome with highly uneven coverage. It was therefore not possible to either assemble the phage from the original sample nor to identify variants at the read level. We believe these results are not conclusive and we prefer not to include them into the manuscript.

REVIEWERS' COMMENTS:

Reviewer #1 (Remarks to the Author):

The authors have carefully revised the manuscript and have answered all questions and queries of the reviewer in a satisfactory way. The report will make an interesting reading for Nature Communication readers interested in microbiology, phage biology, gut virome analysis and phage-bacterium interaction in a mammalian host.
Harald Brüssow

Reviewer #2 (Remarks to the Author):

Comments

I would like to thank the authors for their revisions that have improved significantly their manuscript. I have few comments/remarks mostly to smooth the reading as follows :

In the abstract :

I suggest to move the «not obvious gene for lysogeny » to the sentence after the next one in order to prevent repetition, which should be prevented in abstract.

I would add « during several weeks » at the end of the last sentence.

P2L37 : « the » host

P3L51-52 and 53-54 are basically identical, please correct

P3L64 : is « representative of » more appropriate than representing ?

P3L69 : replace « isolated » by « identified » as isolation suggest that IAS has been cultured while in the same sentence you mentioned that IAS is uncultured.

P5L106 : gp29 and 36 are « putative » structural proteins or « virion-associated proteins », as the RNAPol subunits

P5L108-109 : replace « virion structure » by virion components, or remove « as part of virion structure ».

P5L112 : I'm not familiar with the word « podoviral ». Please stick to the « Podovirus-like » morphology already used in abstract and introduction

P6L147 : replace « have » by « infect »

P7L156 : replace « bacteriophage » by phiCrass001

P8L183 : add « ancestral » bacteriophage. Here the door is open to coevolution studies that will be exciting to conduct.

P9L207 : are you talking about phiCrass001 or Crass-like phage in general

P9L222 : prolonged ?

Point-by-point response to the final reviewers' comments:

Reviewer1: The authors have carefully revised the manuscript and have answered all questions and queries of the reviewer in a satisfactory way. The report will make an interesting reading for Nature Communication readers interested in microbiology, phage biology, gut virome analysis and phage-bacterium interaction in a mammalian host. Harald Brüssow

– No response

Reviewer1: I would like to thank the authors for their revisions that have improved significantly their manuscript. I have few comments/remarks mostly to smooth the reading as follows :

In the abstract :

I suggest to move the «not obvious gene for lysogeny » to the sentence after the next one in order to prevent repetition, which should be prevented in abstract.

– Corrected

I would add « during several weeks » at the end of the last sentence.

– Corrected

P2L37 : « the » host

– Corrected

P3L51-52 and 53-54 are basically identical, please correct

– Corrected

P3L64 : is « representative of » more appropriate than representing ?

– Corrected

P3L69 : replace « isolated » by « identified » as isolation suggest that IAS has been cultured while in the same sentence you mentioned that IAS is uncultured.

– Corrected

P5L106 : gp29 and 36 are « putative » structural proteins or « virion-associated proteins », as the RNAPol subunits

– Corrected

P5L108-109 : replace « virion structure » by virion components, or remove « as part of virion structure ».

– Corrected

P5L112 : I'm not familiar with the word « podoviral ». Please stick to the « Podovirus-like » morphology already used in abstract and introduction

– Corrected

P6L147 : replace « have » by « infect »

– Corrected

P7L156 : replace « bacteriophage » by phiCrass001

– Corrected

P8L183 : add « ancestral » bacteriophage. Here the door is open to coevolution studies that will be exciting to conduct.

– Corrected

P9L207 : are you talking about phiCrass001 or Crass-like phage in general

– The literature evidence is available for the prototypical crAssphage, however, it may exemplify the unusual behaviour of crAss-like phages in general.

P9L222 : prolonged ?

– Corrected